# Inertia Sensors for Measuring Spasticity of the Ankle Plantarflexors Using the Modified Tardieu Scale—A Proof of Concept Study

**DOI:** 10.3390/s22145151

**Published:** 2022-07-09

**Authors:** Megan Banky, Gavin Williams, Rebecca Davey, Oren Tirosh

**Affiliations:** 1School of Medicine, Dentistry and Health Sciences, The University of Melbourne, Melbourne 3010, Australia; megan.banky@epworth.org.au (M.B.); gavin.williams@epworth.org.au (G.W.); 2Physiotherapy Department, Epworth Healthcare, Richmond 3121, Australia; rebecca.davey@epworth.org.au; 3School of Health Sciences, Swinburne University of Technology, Hawthorn 3122, Australia

**Keywords:** spasticity assessment, wearable sensor technologies, inertial measurement unit, Modified Tardieu Scale

## Abstract

Ankle spasticity is clinically assessed using goniometry to measure the angle of muscle reaction during the Modified Tardieu Scale (MTS). The precision of the goniometric method is questionable as the measured angle may not represent when the spastic muscle reaction occurred. This work proposes a method to accurately determine the angle of muscle reaction during the MTS assessment by measuring the maximum angular velocity and the corresponding ankle joint angle, using two affordable inertial sensors. Initially we identified the association between muscle onset and peak joint angular velocity using surface electromyography and an inertial sensor. The maximum foot angular velocity occurred 0.049 and 0.032 s following the spastic muscle reaction for Gastrocnemius and Soleus, respectively. Next, we explored the use of two affordable inertial sensors to identify the angle of muscle reaction using the peak ankle angular velocity. The angle of muscle reaction and the maximum dorsiflexion angle were significantly different for both Gastrocnemius and Soleus MTS tests (*p* = 0.028 and *p* = 0.009, respectively), indicating that the system is able to accurately detect a spastic muscle response before the end of the movement. This work successfully demonstrates how wearable technology can be used in a clinical setting to identify the onset of muscle spasticity and proposes a more accurate method that clinicians can use to measure the angle of muscle reaction during the MTS assessment. Furthermore, the proposed method may provide an opportunity to monitor the degree of spasticity where the direct help of experienced therapists is inaccessible, e.g., in rural or remote areas.

## 1. Introduction

In a clinical context, spasticity is characterised as a ‘velocity-dependent increase in tonic stretch reflexes with exaggerated tendon jerks, resulting from a hyperexcitability of the stretch reflex’ [1]. This suggests that the velocity component of the stretch reflex and the subsequent muscle activation response to fast passive movement are key components in the assessment of spasticity. For quantitative assessment of spasticity, routine clinical scales such as the Modified Ashworth Scale (MAS) and the Modified Tardieu Scale (MTS) are most frequently used [2]. The MTS, however, is better aligned to Lance’s definition of spasticity as it involves two passive movements at a slow joint speed (V1) and a high joint speed (as fast as possible) (V3) [2,3]. The objective of the V3 movement is to generate sufficient joint angular velocity in order to stretch the muscle–tendon complex and activate the spastic muscle response, stimulating a ‘catch’ or ‘clonus’.

During the V3 movement of the MTS, the clinician moves the limb passively as fast as possible, taking the muscle from a shortened to lengthened position. The severity of spasticity is quantified by the clinician in several ways. The assessor grades each V3 movement with a Modified Tardieu Score ranging from no muscle reaction, through to a catch, clonus, and a rigid joint. Additionally, a goniometer is used to measure the joint angle at which a ‘catch’ or clonus occurs during the V3 movement, this is referred to as the angle of muscle reaction (AOR). The difference between the AOR during V3 and the full passive range of motion during V1 is the spasticity angle, with a greater angle indicating a larger degree of spasticity. This technique has several shortfalls, including: (1) goniometer measurements are susceptible to errors as it is hard to identify when a fast movement is suddenly interrupted by a muscle response (resulting in poor accuracy and reliability), and (2) the AOR that is manually identified by the assessor may not represent the true angle at initiation of muscle activation at a physiological level. Previous studies investigating the reliability of a goniometer to measure the AOR during the MTS have demonstrated variable results, particularly when examining inter-rater reliability. For example, Akipinar et al. (2017) reported an ICC (95% CI) of 0.454 (0.22–0.64) for inter-rater reliability of the ankle plantarflexors [4]. Mehrholz et al. (2005) reported an ICC of 0.36 for the soleus muscle and 0.55 for the gastrocnemius muscle [5]. One study reported adequate ICCs for inter-rater reliability of the gastrocnemius (ICC = 0.75) and soleus (ICC = 0.63) [6]. When investigating the validity of the MTS, one study reported 100% agreement between the MTS and EMG in identifying the presence of spasticity; however, a poor relationship between the EMG onset of stretch-induced spasticity and the AOR measured with a goniometer was found (r = −0.57) [7]. Given that costly interventions such as botulinum toxin (BoNT-A) and allied health therapy are often prescribed based on the results of this clinical assessment, there are practical demands for the accuracy and repeatability of current assessment tools to be improved. Recently, the use of sensors was suggested for this purpose [8,9,10].

A newly published systematic review investigated the current use of inertia measuring unit (IMU) sensors to assess spasticity, in studies published between 2017 and 2021 [11]. The review highlights the promising nature of IMUs in this field whilst also identifying several gaps in the current literature requiring further investigation, including: (1) most of the studies have been completed in upper limb muscle groups, (2) the MAS remains the most commonly used clinical spasticity outcome measure, despite this assessment not including a velocity-dependent component, and (3) the MTS was not used in any of the studies included in the review despite the excellent agreement in classification of spasticity between the MTS and EMG [7]. Studies included in the review demonstrated several key findings. The use of IMU sensors combined with surface electromyogram (EMG) to assess elbow flexor spasticity has been demonstrated to be feasible and a suitable method to objectively measure the level of spasticity in a clinical setting [8,9,10]. Zhang et al. [10] placed the IMU at the lower arm to measure the segment angular velocity and placed EMG sensors at the elbow flexors to measure the muscle activation evoked by the passive elbow stretch. Similarly, but without EMG sensors, Kim et al. [9] showed the ability to calculate a meaningful index to quantify spasticity using an IMU placed on the dorsal side of the affected elbow with the use of machine learning methods. A similar IMU/EMG approach was later explored to measure spasticity of the knee extensors in spinal cord injury participants [12]. One IMU was mounted at the shank and the EMG + IMU sensor box on the knee extensor muscle belly. The authors presented a new measure of spasticity termed SPAS that was defined by two parameters a and b that characterised the exponential fit of the spastic torque resulting from involuntary reflexive activation of paralysed muscles derived from the sensor signals. However, none of the included studies combined the MTS with an IMU system; therefore, no simple, clinically feasible, yet accurate approach to identify the AOR during a V3 movement was proposed. Furthermore, none of the studies examined the ankle plantarflexors, which are the most important muscle group for walking [13], have the highest prevalence of lower limb spasticity and are the most frequently injected muscle with BoNT-A [14].

Several earlier studies were found investigating the use of IMUs to detect the AOR during a spasticity assessment. One pilot study explored the use of inertial sensors to measure the joint angle during the Spasticity Test (SPAT) of the Hamstrings, Soleus, and Gastrocnemius in children with cerebral palsy. The authors argued that goniometry is an imprecise method to measure the true AOR in spasticity (maximum 28° error) [15]. They advised to apply inertial sensors when a precise measurement of the AOR is required, for example, when guiding surgical or medical interventions. However, a major limitation of this study is that they did not use EMG to verify that the AOR determined by the IMU system aligned with a true onset of muscle activity. Furthermore, this study used a simplified version of the MTS (i.e., the SPAT). An additional study used IMUs and EMG to measure the AOR in a paediatric cohort with cerebral palsy and reported that muscle activation onset measured by the root-mean-square of the EMG signal occurred very close to the peak angular velocity [16]. Only one study we are aware of comprehensively examined the use of IMUs to assess spasticity of the lower limb using the MTS; however, this was again completed in a paediatric cohort [17]. While these studies have demonstrated promising results and improved the reliability of spasticity assessment, further investigations are required in an adult population and testing speeds should be reported.

Patients with ankle plantarflexor spasticity demonstrate greater gait and balance dysfunction compared to those without this distribution of spasticity [18,19,20,21]. The MTS assessment is often used to measure ankle plantarflexor spasticity and guide treatment planning, but as discussed, the use of IMUs to measure spasticity in adults with plantarflexor spasticity is not well reported. The primary objective of this preliminary study was to explore the use of IMUs in delivering an accurate objective measurement of spasticity, most specifically the AOR, during a MTS assessment of the plantarflexors (Gastrocnemius and Soleus). First, we will explore the relationship between muscle onset (i.e., the AOR) and segment angular velocity measured using EMG and IMU sensors during the MTS assessment. This will follow with an exploratory usability study using two IMUs with no EMG sensors to identify the maximum angular velocity and the AOR during an MTS assessment in people with acquired brain injury. The success of the studies will further inform future studies and will provide affordable possibilities for clinicians to accurately measure lower limb spasticity using the MTS in a clinical setting.

## 2. Materials and Methods

The project included two studies where the first explored the relationship between peak angular velocity and muscle activation onset, and the second explored the use of peak angular velocity to identify the relevant joint angle during MTS. Figure 1 illustrates the schematic diagram of the protocol and measurements.

### 2.1. Study 1: Relationship between Muscle Onset (i.e., AOR) and Segment Angular Velocity

The objective of this study was to identify the association between maximum foot angular velocity and the onset of ankle plantarflexor activation (AOR) during the MTS V3 assessment. The results of this study will provide the rational for the second study in using IMU sensors to accurately quantify the level of spasticity during the MTS V3 assessment.

#### 2.1.1. Participants

Eight participants (m = 5, f = 3, age 44.8 ± 17.3, height 171.5 ± 16.4, weight 70.4 ± 14.5) were recruited for this study. The participants had varied diagnoses including stroke (*n* = 3), traumatic brain injury (*n* = 2), multiple sclerosis (*n* = 1), neuro-oncology (*n* = 1), and lupus with central nervous system (CNS) lesions (*n* = 1). The study was approved by The Alfred Hospital Ethics Committee (project 60/21).

Participants were included in this study if they met the following criteria:Diagnosis of an adult onset acquired neurological condition affecting the CNS;Identified by their treating physiotherapist as having spasticity in their gastrocnemius and soleus, as rated by an X value of ≥ 2;Able to provide informed consent to assessment and cooperate with the testing procedure;Were ≥ 18 years of age;No contraindications to fast passive movement of the affected limb (for example, if non-weight-bearing).

Potential participants were excluded if they had:Severe dystonic movement patterns limiting the completion of the MTS.

#### 2.1.2. Experimental Setup

Three wireless EMG sensors (Noraxon Inc., Scottsdale, AZ, USA) were used to measure muscle activity of the Gastrocnemius Medius (GM), Gastrocnemius Lateralis (GL), and Soleus (SOL). A fourth Noraxon sensor with a built-in tri-axial gyroscope sensor was used to measure the foot angular velocity and was attached to the mid foot (see Figure 2). The wireless EMG sensors consisted of analog sensors (3.4 × 2.4 × 1 cm, 14 g, sample rate up to 4 KHz, baseline noise < 1 µV RMS, CMR > 100 dB), a three-axes accelerometer (+/−200 g), and gyroscope (+/−7000°/s). Following the SENIAM guidelines [22], electrodes were located on the midline of the muscle belly, with the detection surface oriented perpendicular to the long axis of the muscle fibres. Once the electrode position was marked by shaving, gentle abrasion with sandpaper and cleansing with alcohol prepared the skin.

#### 2.1.3. Data Collection

Each participant attended one 30 min session to assess the level of spasticity of the Gastrocnemius and Soleus using the MTS assessment. The assessor had more than seven years clinical experience and routinely performed this assessment. The patient remained in a relaxed, supine position on a therapy plinth with their knee at 0 degrees for Gastrocnemius and at ~45 degrees flexion for Soleus trials (see Figure 2). Three trials were performed for Gastrocnemius and three for Soleus. The clinician completed the MTS as per the standardised protocol: “V1 = slow, passive movement” and “V3 = as fast as possible”. Only the V3 trials were analysed for this study. Data was sampled (2000 Hz) and recorded using the Noraxon myoRESEARCH software and exported to c3d file format for further processing and analysis in Visual3D (c-motion Inc., Kingston, ON, Canada).

#### 2.1.4. Signal Processing and Outcome Measures

The sEMG were first bandpass filtered (20–500 Hz). To generate the linear envelope (LE) of the signal, the bandpass filtered EMG signal was rectified and low passed filtered at 6 Hz using a zero-lag fourth-order Butterworth filter. Onset of muscle activation (i.e., the AOR during V3) was visually extracted at the initiation of abrupt rise in the LE signal of each muscle. Outcome measures included: (1) the time delay between muscle activation onset and the time of maximum foot angular velocity (Δt) (see Figure 3) and (2) maximum foot angular velocity (ω_max_).

### 2.2. Study 2: The Use of IMUs to Measure the Angle of Muscle Reaction during the V3 MTS Assessment

#### 2.2.1. Participants

The study was approved by The Alfred Hospital Ethics Committee (project 60/21). The inclusion and exclusion criteria were identical to study 1. Twenty participants (m = 11, f = 9, age 46.55 ± 16.79, height 170.31 ± 9.23, weight 76.10 ± 14.39) were recruited for this study. The participants had varied diagnoses, including stroke (*n* = 8), traumatic brain injury (*n* = 5), multiple sclerosis (*n* = 4), and other neurological conditions such as neuro-oncology (*n* = 3).

#### 2.2.2. Experimental Setup

Two small, clinically feasible, skin-mounted Yost Labs 2-SpaceTM Wireless IMU Sensors (35 × 60 × 15 mm, 28 g, 2.4 GHz, range 50 m) were used to collect the tri-axial acceleration (8 g, sensitivity 0.00024 g/digit–0.00096 g/digit) and angular velocity (gyro scale 500°/s, sensitivity 0.00833°/s) of the lower leg and foot during the MTS spasticity assessment. The foot IMU was strapped around the foot using a Velcro strap while the other IMU was laced on the flat surface on which the shank was resting on, as illustrated in Figure 4.

#### 2.2.3. Data Collection

Data collection was similar to the study 1 protocol; however, for the Soleus trials the knee was placed in approximately 90° flexion as per the standardised MTS protocol [23] (this was not feasible in study 1 due to the EMG placement). The therapist completed the MTS as described earlier for both Gastrocnemius and Soleus trials. A custom-made Python (version 2.7 https://www.python.org/, accessed on 18 March 2021) data processing program was developed to record the acceleration and angular velocity from the two IMUs sensors at 100 Hz sampling frequency. The sensors were programmed to connect to the software via the 3-SpaceTM Wireless Dongle. The software provided real-time feedback regarding the peak ankle dorsiflexion (DF) angular velocity and the ankle angle during each trial. Additionally, following each completed trial the data was exported and saved as a comma separated CSV file for further analysis.

#### 2.2.4. Signal Processing and Outcome Measures

The foot angular velocity was measured from the foot IMU using the gyroscope as described in study 1. Ankle joint angle was calculated from the differences between the IMU orientation angles presenting the foot and shank segments. Segment orientation angles were calculated from the acceleration and angular velocity signals as described in Abhayasinghe et al. [24]. The authors validated a computational orientation estimation algorithm (Gyro Integration-Based Orientation Filter—GIOF) that was used to estimate the forward and backward swing angle of the thigh (thigh angle) for a vision-impaired navigation aid. The maximum foot angular velocity (ω_max_), angle at maximum angular velocity (θ@max), the maximum DF angle (θ), and the differences (Δθ) between θ@max and θ, were extracted from the IMU-processed acceleration and angular velocities (see Figure 5). The time of the ω_max_ was considered as the AOR (i.e., the muscle activation or onset of the ‘catch’ or ‘clonus’) due to the negligible delay between AOR and muscle activation identified in study 1.

## 3. Results

### 3.1. Study 1

Table 1 shows the mean ± standard deviation and range of each muscle for the time delay between muscle activation onset and maximum foot angular velocity (Δt), and the maximum angular velocity of the foot (ω_max_) during MTS Gastrocnemius and Soleus trials. For the Soleus trials only the Soleus data is presented as during the soleus trial the body position of having a flexed knee reduced the Gastrocnemius activity. The average Δt for both the medial and lateral heads of the Gastrocnemius muscle was 0.049 s which was very close to the soleus with 0.046 s delay. The peak foot angular velocities on average were 452.5°/s and 558.6°/s for Gastrocnemius and Soleus, respectively.

### 3.2. Study 2

Table 2 shows the means ± standard deviation and range of the maximum angular velocity (ω_max_), angle at maximum foot angular velocity (θ@max), angle at maximum DF (θ), and the differences (Δθ) between θ@max and θ during MTS Gastrocnemius and Soleus trials. All outcome measures were similar between Gastrocnemius and Soleus MTS tests. The measured angle at maximum velocity (θ@max) was significantly greater compared to the angle measured at maximum DF (θ) for both Gastrocnemius and Soleus MTS tests (*p* = 0.028 and *p* = 0.009, respectively).

## 4. Discussion

Study 1 found that the maximum foot angular velocity occurred 0.049 s and 0.032 s following the spastic muscle reaction for Gastrocnemius and Soleus, respectively. The use of the peak angular velocity, therefore, is suggested as an accurate method to identify the AOR when EMG sensors are not available. This system has greater reliability than a goniometer and is more clinically feasible than an EMG-based assessment, making it a practical assessment modality to implement in a clinical setting. Bar-On et al. [25] also showed that muscle activation onset measured by the root-mean-square of the EMG signal occurred very close to the peak angular velocity. While EMG lacks clinical utility and is not feasible in routine clinical practice due to the specialised training and equipment required, more accessible technologies such as IMUs or Smartphones are able to accurately measure joint angular velocity and AOR, as demonstrated in study 2. As such, the use of IMUs may assist clinicians in accurately measuring spasticity using the MTS, improving the validity of the scale in clinical practice.

The first study reported in this paper was performed as a response to a key limitation within the aforementioned van den Noort study [15]. They suggested that a precise spasticity assessment should involve both electrophysiological and biomechanical aspects to identify the AOR and its association to joint kinematics. In study 1, we used EMG sensors to measure electrophysiological signals and an IMU sensor to measure angular velocity (biomechanical aspects), with promising results. Measuring electrophysiological signals, however, requires specialised and costly sensors, something that is not feasible to implement in everyday clinical practice. Comparatively, user friendly, affordable, and wearable devices continue to gain traction as they allow for an effective and efficient assessment with minimal user training required.

Our second study, therefore, explored a more clinically feasible option with the use of two affordable IMU sensors to identify the AOR during the MTS, without EMG sensors. Using these sensors, we were able to accurately determine the peak angular velocity and subsequent AOR. This is likely to have greater reliability and validity than the current clinical method of using a goniometer, which has been shown to have poor reliability and validity compared to a criterion reference three-dimensional motion analysis system when using the MTS to assess for spasticity of the Gastrocnemius and Soleus [26,27]. Furthermore, the method used in study 2 can be supported by the electrophysiological approach investigated within study 1, compared to a goniometer, which relies on the objectivity of the assessor and where in the movement they feel the muscle reaction. It is well established that the MTS is the most appropriate measure to assess for spasticity in a clinical setting; therefore, this approach has the potential to improve the reliability of the scale and assist in guiding treatment planning.

Previous literature investigating the use of IMUs to assess spasticity has demonstrated promising results in terms of improving assessment reliability [3,11,15,17,25]. However, the clinical utility and ease of implementation for these previously reported systems is variable. This study proposes the first method of assessing ankle plantarflexor spasticity, in an adult cohort using the MTS and an IMU (USD 600) system. The proposed system validated with EMG (study 1) is affordable, requires minimal training, and is able to accurately report testing speed and the AOR, using the angle at peak velocity. Due to its reliability, as well as the ease of incorporating it into a clinical setting, it has the potential to improve current clinical assessment of spasticity, guide treatment planning, and monitor progress. Furthermore, it can be easily implemented into all settings including those in rural or remote areas where access to complex equipment may be limited.

Currently, there is mixed evidence regarding the impact of spasticity interventions such as BoNT-A on patient outcomes. One possible explanation for this is the poor reliability and validity of current clinical assessment tools [28,29]. This present study addresses this issue, not by creating a new assessment tool but instead by improving the ability to measure the AOR and subsequent spasticity angle of an existing tool. Additional research is required to investigate whether controlling testing speed, using immediate feedback provided by this IMU system, impacts the assessment findings when using the MTS [30]. Standardising testing speed and matching this speed to relevant joint angular velocities during functional tasks, such as walking, may improve the ecological validity of the MTS [30]. This may improve clinicians’ ability to select patients who would benefit from active interventions such as BoNT-A, resulting in improved patient outcomes and reduced healthcare wastage. The IMU system used in this study is able to provide real-time feedback regarding testing speed and may be used to further standardise this component of the assessment.

In summary, the IMU system trialled in this study was found to accurately calculate the AOR using the angle at peak velocity (when compared to EMG muscle onset) and successfully measure and provide real-time feedback regarding joint angles and joint angular velocity. This eliminates the large error associated with goniometric measurements currently adopted in daily clinical practice. This technology may therefore assist in accurate measurement, and subsequent clinical decision making, including guiding treatment paradigms such as rehabilitation programs, BoNT-A administration, and surgical interventions.

### Limitations

This was a preliminary feasibility and proof of concept study designed to investigate whether peak joint angular velocity, measured using an IMU system, could accurately detect and measure the AOR during a spasticity assessment using the MTS. While the results were promising, there were several limitations of the study design and subsequently, areas of further research required. Firstly, the sample size in study 1 was small, yet adequate to demonstrate the accuracy of the method. Furthermore, only one assessor completed the assessment, at one time point. Additional research is now required with larger sample sizes to investigate repeatability, reproducibility, and quantify the difference between this novel approach and traditional assessment methods. These studies are likely to justify the use of wearable sensors in a clinical setting to assist in monitoring progress and guiding intervention.

## 5. Conclusions

When using an EMG and IMU measuring system, study 1 demonstrated that the maximum foot angular velocity occurred 0.049 and 0.032 s following the spastic muscle reaction for Gastrocnemius and Soleus, respectively. This key finding confirms that the use of peak angular velocity is a good indication of spastic muscle onset. In the absence of equipment and personnel trained in the use of EMG devices, study 2 successfully demonstrated that clinically feasible, wearable technology can be used to identify the onset of a spastic muscle reaction. As such, this work proposes a more accurate, clinically feasible method than the currently used goniometer, to quantify the AOR during a spasticity assessment using the MTS.

## Figures and Tables

**Figure 1 sensors-22-05151-f001:**
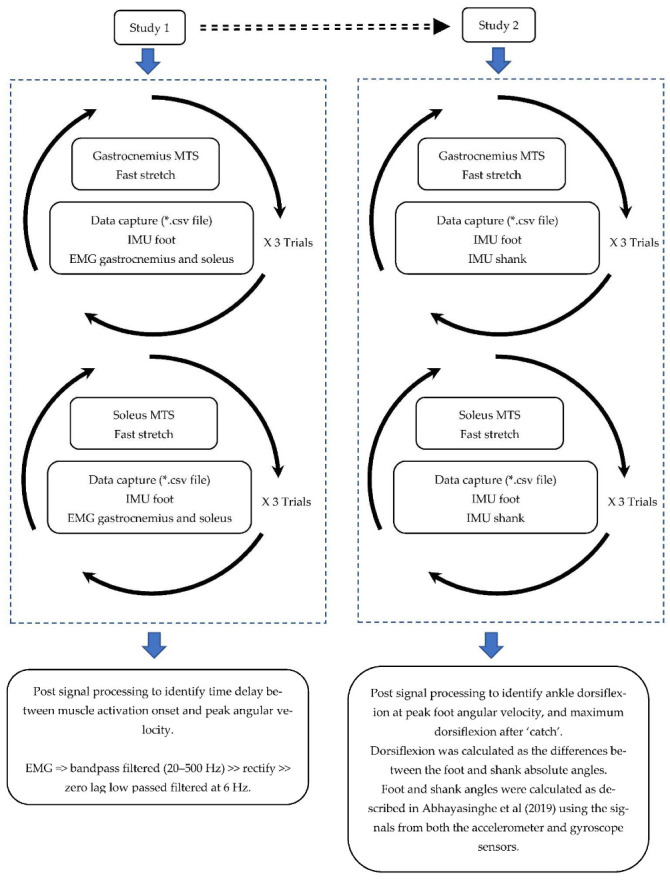
Schematic diagram of the MTS assessment and data processing in studies 1 and 2.

**Figure 2 sensors-22-05151-f002:**
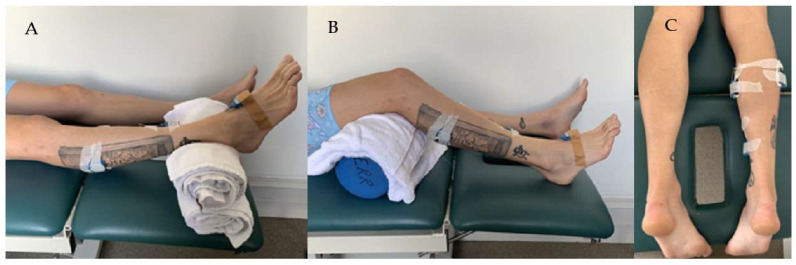
Participant position and sensor setup during the Tardieu spasticity assessment of the Gastrocnemius (**A**) and Soleus (**B**). EMG sensors positions are illustrated in (**C**).

**Figure 3 sensors-22-05151-f003:**
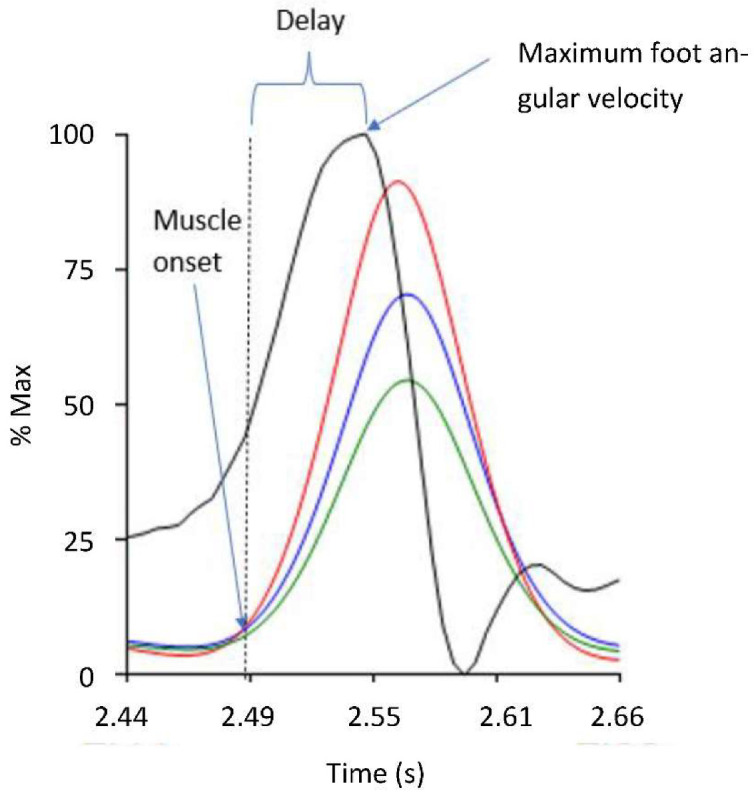
The delay (Δt) from the muscle activation onset (i.e., AOR) and the maximum foot angular velocity. The vertical axis represents the percentage of maximum angular velocity (black) and linear envelope of the Gastrocnemius Medius (red), Gastrocnemius Lateralis (blue), and Soleus (green) muscles activations during the MTS V3 test, normalised to their peak value (% of max).

**Figure 4 sensors-22-05151-f004:**
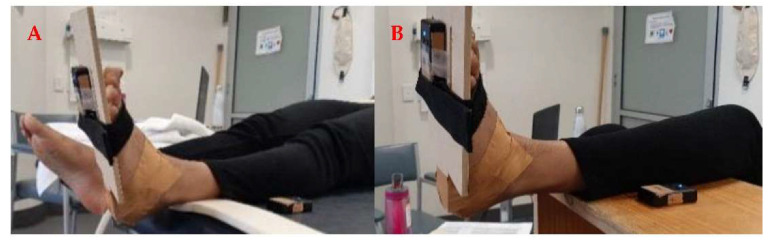
Participant position and sensor setup during the Tardieu spasticity assessment of the Gastrocnemius (**A**) and Soleus (**B**).

**Figure 5 sensors-22-05151-f005:**
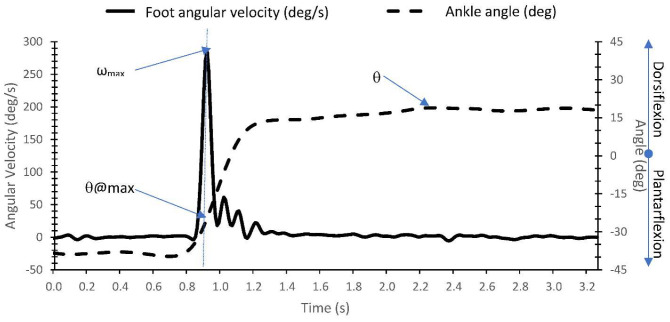
Identification of maximum foot angular velocity (ω_max_), angle at maximum angular velocity (θ@max), and angle at maximum DF (θ) from the IMU acceleration and angular velocity measurements.

**Table 1 sensors-22-05151-t001:** Means standard deviation (range) of the delay (Δt in seconds) from the Gastrocnemius medius, Gastrocnemius lateralis, and Soleus activation onset and maximum foot angular velocity during the Gastrocnemius and Soleus MTS assessments.

MTS Test	Med Gastroc Δt (s)	Lat Gastroc Δt (s)	Soleus Δt (s)	Angular Velocity ω_max_ (°/s)
Gastrocnemius	0.049 ± 0.022 (0.020–0.080)	0.049 ± 0.020 (0.010–0.080)	0.032 ± 0.075 (0.020–0.120)	452.5 ± 70.9 (325–608)
Soleus			0.046 ± 0.026 (0.020–0.100)	558.6 ± 72.8 (440–726)

**Table 2 sensors-22-05151-t002:** Means ± standard deviation (range) of the maximum foot angular velocity (ω_max_), angle at maximum angular velocity (θ@max), angle at maximum DF (θ), and the differences (Δθ) between θ@max and θfrom during Gastrocnemius and Soleus Tardieu tests (MTS).

MTS Test	ω_max (deg/sec)_	θ@max _(deg)_	θ _(deg)_	Δθ _(deg)_
Gastrocnemius	430.3 ± 90.4 (262.7–612.4)	−16.1 ± 8.7 (−35.0–2.8)	12.3 ± 8.3 (−1.0–34.7)	28.4 ± 10.9 (14.7–55.9)
Soleus	439.9 ± 74.5 (260.4–575.6)	−10.9 ± 8.3 (−24.6–2.5)	17.4 ± 8.6 (1.2–33.9)	28.2 ± 11.4 (10.6–47.5)

## Data Availability

Data available on request due to restrictions. The data presented in this study are available on request from the corresponding author. The data are not publicly available due to participants privacy.

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
