# Peer review of "Inertia Sensors for Measuring Spasticity of the Ankle Plantarflexors Using the Modified Tardieu Scale—A Proof of Concept Study"

_sensors, 2022, doi:10.3390/s22145151_

Round 1

Reviewer 1 Report

Title: Inertia sensors for measuring spasticity of the ankle plantar- flexors using the Modified Tardieu Scale

Journal: Sensors (ISSN 1424-8220)

Manuscript ID: sensors-1784024

Authors: Banky et al.

In this manuscript, authors proposed an accurate method to accurately determine the ankle angle during the Modified Tardieu Scale test by measuring  the maximum angular velocity and the ankle angle, at the maximum angular velocity, using two affordable inertial sensors.

1.       Kindly modify the abstract section, the findings of this study should be clearly stated in the abstract section. Findings means numbers.

2.       Figure 2 and Figure 4 are not consistent. Kindly, modify both figures so that the font size and type for x-axis and y-axis labels and tiles  are  similar

3.       The manuscript includes so many abbreviations such as MTS, @max, @...etc. Kindly create an abbreviation table and identify each abbreviation.

4.       A schematic for measurement method should be created and included in the manuscript as a new figure. This is very important for understanding the way how findings of this manuscript was found.

5.       In the experimental section for study 1 and 2, technical specification of all sensors (wireless EMG sensors (Noraxon Inc) and IMU sensors … ) should be indicated.

6.       The finding of this study should be compared to other reported work. If possible, create a table and compare results in this manuscript with other reported work. Also, include more references.

7.       The conclusion section is not well written and needs a modification. Kindly re-write the conclusion and avoid words such as These findings  and this work. Instead state those findings and make the conclusion section understandable. 

Reviewer 2 Report

With the development of IMU and wearable sensor technology has become a widespread and useful tool in clinical practice. Authors presents a new method to determine the ankle angle during the MTS test by measuring the maximum angular velocity and the ankle angle, at the maximum angular velocity, using two affordable inertial sensors

Authors didn’t explain how their work differs from previous work and what are the highlights: Which novelty do authors claim for this review article? This IMU based Range of Motion assessment is a widely used approach. The aim of this manuscript is very frequent and a similar article has been published in this journal.

Sample size don’t supports the conclusions drawn by the author and also the number of the participans is lower than the articles in the literature on this topic.

I suggest the authors to increase the number of patients and add tests on the repeatability and reproducibility of this new test also by comparing different practitioners.

I also suggest starting from a more careful analysis of literature by inserting articles such as:

Weizman, Yehuda, et al. "Recent State of Wearable IMU Sensors Use in People Living with Spasticity: A Systematic Review." Sensors 22.5 (2022): 1791.

Oliva-Lozano, José M., Isabel Martín-Fuentes, and José M. Muyor. "Validity and reliability of an inertial device for measuring dynamic weight-bearing ankle dorsiflexion." Sensors 20.2 (2020): 399.)

Reviewer 3 Report

Recommendation: Publish after major revisions noted.  

Comments:  

This manuscript demonstrates how wearable technology can be used in a clinical setting to identify the onset of muscle spasticity and proposes a more accurate method that clinicians can use to measure the angle of muscle reaction during the MTS assessment. The authors need to address the following comments and revise the manuscript accordingly. 

  1. Please consider to highlight the technology in determining the effect of interventions, including rehabilitation programs, botulinum toxin injections, and orthopedic surgeries.
  2. Provide a table and show comparison of MTS (spasticity) assessments using IMU.
  3. Add a schematic diagram of the MTS assessment.
  4. To calculate the joint angle, each segment angle should be obtained. Provide the equation. 
  5. Discuss how to improve the reliability of spasticity assessment. In addition to the joint angle calculation algorithm, discuss a muscle reaction detection method as well as a visual biofeedback mechanism to improve the reliability. Consider the velocity-dependent characteristics of muscle reaction. 

Round 2

Reviewer 2 Report

paper has been improved

Reviewer 3 Report

The authors have addressed the comments quite thoroughly and this version of the manuscript is improved. Please publish.